# Investigation on the Interface Morphology of Mg/Al Corrugated Composite Plate in the Straightening Process

**DOI:** 10.3390/ma15134383

**Published:** 2022-06-21

**Authors:** Tong Xing, Cuirong Liu, Jie Liu, Hailian Gui, Xiaotong Hu, Zhibing Chu, Leifeng Tuo

**Affiliations:** 1Engineering Research Center of Heavy Mechanical, Ministry of Education, Taiyuan University of Science and Technology, Taiyuan 030024, China; b20204791002@stu.tyust.edu.com; 2Material Science and Engineering College, Taiyuan University of Science and Technology, Taiyuan 030024, China; liuruirong@tyust.edu.cn (C.L.); b20214310013@stu.tyust.edu.com (J.L.); s202114210065@stu.tyust.edu.com (X.H.); 2011006@tyust.edu.cn (Z.C.); 2021021@tyust.edu.cn (L.T.)

**Keywords:** Mg/Al corrugated composite, interface morphology, straightening, finite element analysis

## Abstract

The Mg/Al composite plate was developed in aerospace and other fields. At present, through the corrugated rolling method, the bonding strength of Mg/Al composite plate could be increased and the warpage could be reduced. However, this still requires the straightening process to reach the parameters’ range. In this work, the original interface morphology of Mg/Al corrugated composite plate was obtained by experimental characterization. Based on the principle of elastoplastic mechanics, the equations of straightener parameters and straightening process parameters were obtained and the influencing factors were deduced. So, the straightening model was established in an Abaqus. The effects of straightener parameters and straightening process parameters on the interface morphology were analyzed and the interface morphology was expressed by amplitude and period length of the equation. The results showed that bending moment, shear strength and the reduction of second roll played roles on the interface morphology. After the first straightening unit, the amplitude increased by 1.1% and the period length increased by 3%. Finally, a complete straightening parameter was designed, which included straightener parameters, straightening process parameters and straightening temperature. The aim of this work was to provide a theoretical basis for establishing a high precision Mg/Al corrugated composite plate straightening model, which could improve the bonding strength while ensuring the straightening effect.

## 1. Introduction

As a new type of composite plate material, Mg/Al composite plate has been widely used in modern industrial production (such as aerospace engineering and electronic fields), attributed to light weight and high specific strength from Mg alloy, as well as corrosion resistance and low price from Al alloy [1,2,3]. The production methods of Mg/Al composite plate have been studied, such as traditional hot rolling [4], porthole die coextrusion and forging (PCE-F) [5], and hard-plate rolling (HPR) [6]. In the traditional hot rolling method, the bonding strength decreased with reduction ratio and rolling temperature due to the increase in the grain size and diffusion layer width with the reduction ratio and rolling temperature [4]. The first stress-drop was determined by IMCs (intermetallic compounds) and microstructures of the Mg layer, while the second stress-drop was determined by IMCs and microstructures of the Al layer during the tensile test of Al/Mg/Al plate [5]. Compared to traditional hot rolling, the thickness of the diffusion layer could reach 38 µm under the condition of 60% reduction, the yield strength could be 172.3 MPa and the elongation could be 21.5% by HPR. At the Al/Mg interface, two clear layers of Al_3_Mg_2_ and Al_17_Mg_12_ intermetallic compounds appeared and thickness of diffusion layer was uniform [6]. Therefore, the new phases of Al_17_Mg_12_ [7] and Al_3_Mg_2_ [8] formed at the Mg/Al interface was verified. Plate shape warpage and low bonding strength would inevitably appear in the traditional rolling method due to the great difference in ductility between Mg and Al [9]. Therefore, corrugated rolling has been developed to solve these problems. The bonding strength would be enhanced, owing to the interface extension of the Mg/Al composite layer by corrugated rolling [10,11,12]. Moreover, the interface force is less than that produced by flat-roll rolling, which is caused by the angle between the vertical stress and the bonding interface, so that the service strength and life of Mg/Al corrugated composite plate would be significantly increased [13]. After corrugated rolling, the straightening process is essential to achieve the strict shape requirements and specific residual stress range in extreme service conditions of high temperature and etching [14,15]. The elastic–plastic deformation at the interface of Mg/Al corrugated composite plate in straightening would have a great impact on the interface morphology [16]. In the rolling, a mathematical model which reflected the change of the maximal absolute shear stress for the center layer was established, by which the maximal absolute shear stress for the center layer can be easily calculated [17]. Meanwhile, the roles of strain and state-of-strain on particle fracture and grain size control have been developed to be applied to Mg alloy and Al alloy [18,19]. Therefore, it is of great significance to determine the parameters of the straightener and straightening process and explore the variation law of corrugated interface morphology [20,21,22].

Recently, many researchers have analyzed the influence of straightener parameters on straightening effect. The relationships between editing power spent on plate alternating bending, power spent for friction of rollers on surface in rolling motion, power spent for the overcoming of frictional force on necks of carrying rollers and different straightening machine diameters were calculated [23]. To straighten model in FEM, the accuracy of results was also determined by the definition of the materials. An incompressible, isotropic, nonlinear elastic rectangular block and a circular cylindrical sector were set as the definitions of the materials. Afterwards, the straightening process defined by these materials was simulated, respectively, in Abaqus [24]. By optimizing the structure and parameters of the straightener, the labor intensity of workers would be reduced, production efficiency would be improved and product quality and qualified rate of products would be greatly increased [25]. The straightening effect was affected not only by straightening parameters but also by straightening process parameters. The section bending characteristics of bimetal composite plates in straightening were analyzed by the curvature integral method. It indicated that the neutral layer of Al/steel composite plates appeared in straightening and residual stress was small after straightening [26,27]. The process of single bending, springback deformation and residual deformation of the steel tube were deduced in eleven cross-roll tube straightening, and the final residual curvature of the microbeam segment was calculated through iterations repeating bending and springback to improve straightening accuracy [28,29].

In our previous work, the straightening of the S304/Q235 composite plate and Mg alloy was analyzed by the boundary finite element method and the elastic–plastic method [30,31,32,33,34]. On the one hand, the influences of neutral layer offset and Bauschinger effect on straightening accuracy were explored in straightening. The neutral layer offset and Bauschinger formulas were deduced to obtain the straightening force formula, and the influencing factors were discussed. On the other hand, the influence of straightener parameters on the effect was studied and straightening parameters were determined by the elastic–plastic method and the curvature integration method. In addition, the straightening of Mg alloy was studied, and the results showed that tension–compression asymmetry of AZ31B Mg alloy and neutral layer offset had played an important role in straightening accuracy.

In this work, the physical properties of Mg/Al corrugated composite plate were determined by XRD, and the interface morphology parameters of Mg/Al corrugated composite plate were determined by EDS and an ultra-depth-of-field microscope, which provided experimental data for establishing a finite element model for the straightening process of Mg/Al corrugated composite plate. According to the calculation formula of straightener parameters and straightening process parameters, the process parameters of Mg/Al corrugated composite plate were determined. The straightening process of Mg/Al corrugated composite plate was established by Abaqus finite element software and the influence on the corrugated interface was discussed through different straightener parameters and straightening processes. The aim of this work was to establish an appropriate straightening model while ensuring the straightening effect, so as to increase the surface area of the composite plate and improve the bonding strength of the plate. It could provide a theoretical basis and experimental support for the straightening model of corrugated composite plate.

## 2. Determination of Corrugated Interface of Mg/Al Composite Plate after Rolling

The rolling and straightening process flow chart of Mg/Al corrugated composite plate is shown in Figure 1. Firstly, an AZ31B Mg alloy sheet was welded with a 5052 Al alloy sheet. Then, they were rolled to a compound by corrugated roll and flat roll. The rolling parameters are shown in Table 1 and the rolled plate is shown in Figure 2. After that, the Mg/Al corrugated composite plate was turned over and sent to the eleven-roll straightener for straightening. Finally, the Mg/Al composite plate with a flat surface and a corrugated bonding interface was obtained.

The samples of Mg/Al corrugated composite plate were prepared at the size of 10 mm × 10 mm × 4 mm. The phase structure of the interface of the Mg/Al corrugated composite plate was analyzed by X-ray diffractometer (XRD, DX2700, Dandong, Liaoning, China) and the increment step was set as 4°/min, as shown in Figure 3a. A scanning electron microscope (SEM, JSM-7001F, Japan Electronics Co., Ltd., Tokyo, Japan) and an energy dispersive spectrometer (EDS) were used to observe the surface micromorphology of the Mg/Al corrugated composite plate. According to the X-ray wavelength difference of Mg and Al elements, the types and contents of Mg and Al elements on the material surface were determined, as shown in Figure 3b,c. The interface of the Mg/Al corrugated composite plate was observed by an ultra-depth-of-field microscope (VHX-1000, Keyence, Osaka, Japan), as shown in Figure 3d.

In Figure 3a, according to the XRD result, new phases (Al_12_Mg_17_ and Al_3_Mg_2_) were produced at the interface of the Mg/Al corrugated composite. The plane indices of Al_12_Mg_17_ were (6 6 2), (7 2 1) and (6 5 1), and the plane indexes of Al_3_Mg_2_ were (12 12 10). In Figure 3b,c, the interface thickness of the Mg/Al corrugated composite plate was 20 μm after two passes of hot rolling. Therefore, when the finite element model was established, the Mg/Al corrugated composite interface bonding layer was divided into an Al_12_Mg_17_ layer and an Al_3_Mg_2_ layer, and the thickness of each layer was 10 μm, respectively. In Figure 3d, according to the interface morphology photographed by the ultra-depth-of-field microscope, the waveform equation of the corrugated interface morphology was extracted by GetData software as $y=0.1\sin\frac{\pi }{5}x$.

## 3. Determination of the Straightening Machine Parameters and Straightening Process Parameters of Mg/Al Corrugated Composite Plate

The straightening effect of Mg/Al corrugated composite plate depends on straightening machine parameters and straightening parameters, and different parameters show different effects on the interface of Mg/Al corrugated composite plate.

### 3.1. Roll Distance and Roll Diameter of the Straightening Roll

The maximum reduction of straightening roll inlet roll is:(1)$$\Delta =\frac{{2\sigma }_{s}p^{2}}{2.7\mathrm{EH}}$$
where $\sigma_{s}$ is the yield strength of the straightening part (material), p is the roll distance of the straightening roll, E is the elastic modulus of the straightening part and H is the thickness of the straightening part.

To ensure that the straightening part is bitten by the straightening roll, the mechanical conditions are as follows:(2)$$2\mu \cos\alpha +\left(\mu^{2}-1\right)\sin\alpha \;\geq \;0$$
where μ is the friction coefficient between the straightening part and the straightening roller, α is the bite angle of straighten part, $\cos\alpha =\frac{R-\Delta }{R}$ and $\sin\alpha =\frac{\sqrt{2R\Delta -\Delta^{2}}}{R}$.

When μ = 0.3, the maximum value of roll distance can be obtained as follows:(3)$$p_{\max}=\frac{2\mathrm{EH}}{4{.45\sigma }_{s}}$$

The maximum value of roll diameter is:(4)$$D_{\max}{=2\alpha p}_{\max}$$

The minimum value of roll diameter is:(5)$$D_{\min}=0.836\sqrt{\frac{{\alpha Z}_{t}{\mathrm{EM}}_{\max}}{\sigma_{t}}}$$

$M_{\max}$ is plastic’s ultimate bending moment. $Z_{t}$ is section modulus. For rectangular sections, $M_{\max}=1.5$ and $Z_{t}=\frac{{\mathrm{BH}}^{2}}{6}$.

The minimum value of roll distance is:(6)$$p_{\min}=\frac{D_{\min}}{2\alpha }$$

### 3.2. Roll Length of the Straightening Roll

The roll length of straightening roll:(7)$${L=B}_{\max}{+\alpha }_{1}$$
where $B_{\max}$ is the straightening width and $\alpha_{1}$ is the allowance at both ends of the straightening part and the straightening roller. When $B_{\max}\;<\;200\;\mathrm{mm}$, $\alpha_{1\;}=50\;\mathrm{mm}$, and when $B_{\max\;}>\;200\;\mathrm{mm}$, $\alpha_{1}=100\sim 300\;\mathrm{mm}$.

### 3.3. Roll Speed of the Straightening Roll

The roll speed of straightening roll:(8)$$v=\frac{R\eta N}{\left(T_{\varepsilon }{+T}_{m}\right)}$$
where R is the radius of the straightening roll, η is the transmission efficiency of the straightening roll, N is the driving power of the straightening roll, $T_{\varepsilon }$ is the total shaping bending torque of the straightener and $T_{m}$ is the rolling and friction torque of the straightening roll.

### 3.4. The Straightening Process Parameters

The reduction of the tenth roll at the outlet of the eleventh roll straightening is relatively small, so that the deformation of the composite plate can be considered to have elastic deformation or no deformation. Therefore, the reduction of the tenth roll is:(9)$$\delta_{10}=\left(0\sim 1\right)\frac{\sigma_{s}l^{2}}{12\mathrm{Eh}}\;$$

The reduction of the second roll is:(10)$$\delta_{2}=\frac{8}{7}\times \frac{\sigma_{s}1^{2}}{12Eh}\left(K-K\prime \right)-\frac{1}{7}\delta_{10}$$
where
$K_{\prime }=\frac{24\mathrm{Eh}\nu_{0}}{\sigma_{s}1^{2}}$, $K_{1}=\frac{5}{{2C}_{y}^{2}}+\frac{E}{E_{1}}\left(\frac{C_{y}}{2}-\frac{27}{{16C}_{y}^{2}}\right)+\frac{9}{8}\left[1\;-\;\frac{E}{E_{1}}\right]\left(1\;-\frac{9}{{4C}_{y}^{2}}\right)+\frac{{3C}_{z}-{\;C}_{y}}{4}$, $\nu_{0}$ is deflection caused by original curvature, $C_{z}$ is relative total deformation curvature and Cz= h2z0, $z_{0}$ is elastic–plastic bending deformation degree and z0=σsE1ρz, and $C_{y}$ is relative elastic complex curvature and $C_{y}=\left(\frac{3}{2}-\frac{0.5}{C_{z}^{2}}\right)\left(1\;-\frac{E_{1}}{E}\right)+\frac{E_{1}}{E}C_{z}$. The relative value is the ratio of each curvature and yield curvature 1ρw. $E_{1}$ is work-hardening modulus.

The reduction of the second roll of the eleven-roll straightener $\delta_{2}$ and the reduction of the tenth roll $\delta_{10}$ are determined. According to the overall tilt rate straightening scheme, the reduction of all straightening rollers can be calculated.

According to the equations of the straightening parameters in Equations (1)–(8), roll distance, roll diameter, roll speed and straightening parameters of the straightening roll have an important impact on the straightening corrugated interface, which needs to be deeply discussed. Furthermore, the effects of the stress distribution of the width of the composite plate is mainly affected by the roll length. Generally, the width of Mg/Al corrugated composite plate is far less than the roll length of the straightening roll in the straightening process, so the later simulation is not discussed here.

## 4. Establishment of the Straightening Model of Mg/Al Corrugated Composite Plate

The thermal straightening process model of Mg/Al corrugated composite plate was established by Abaqus finite element software, as shown in Figure 4. Figure 5 showed a single straightening unit model. The straightening temperature was set at 300 °C and the original curvature of the composite plate was set to 0.

The eleven-roll straightener was applied as a prototype and an Abaqus model of the straightening roll in the straightener was established. A full hydraulic screwdown system was employed in this straightener, which could accurately adjust the bending amount of the straightening roll. The general parameters of the straightener are shown in Table 2 and the specific values of reduction are shown in Table 3.

A bimetallic composite plate was selected as the straightening simulation plate. The base material was 5052 Al alloy, and the cladding material was AZ31B Mg alloy. The length of the composite plate was 500 mm, the width was 30 mm and the total thickness was 4 mm.

Four layers of composite materials with thickness were established. The cladding layer was 1.49 mm, the base layer was 2.49 mm and the transition layer was 0.02 mm. The material properties are shown in Table 4. The material model was set as isotropic elastic–plastic material and the work-hardening modulus was taken as the empirical value of 0.01E.

In this work, the stress-and-strain state of the bonding layer were mainly analyzed and the force on the straightening roll was small in the single straightening process, so the straightening roll could be defined as an analytical rigid body. All-round displacement constraints and X and Y steering constraints were imposed on the straightening roller in the model because the straightening roller only rotated circumferentially along the center line. Meanwhile, a zero-displacement constraint in the X and Y directions were imposed on the left side, and a zero-displacement constraint in the Y direction was imposed on the right side in the single straightening unit. It could simulate the metal fluidity of the plate in the straightening. The composite plate material was meshed by HyperMesh software and imported into Abaqus, as shown in Figure 6. The material property of the composite plate was set as isotropy. The contact between the composite plate and the straightening roller was set as surface to surface, and the dynamic friction coefficient was 0.25.

The eleven-roll straightener was used to simulate the influence of straightening process parameters and temperature on interface morphology. In this part, the straightening process was divided into two steps. The first step was that the plate was fed into the straightener and the time was 0.1 s. The second step was the straightening process and the time was 19.2 s. A single straightening unit model was used to simulate the influence of straightener parameters on interface morphology. In this part, the influences of holding time and springback is not considered.

## 5. Results

The interface of Mg/Al corrugated composite plate was approximately regarded as a sinusoidal equation. So, the interface morphology was determined by amplitude and period length of the equation interface morphology. The period length characteristic curve with straightening time is shown in Figure 7 and the amplitude characteristic curve with straightening time is shown in Figure 8.

In Figure 7, the straightening of Mg/Al corrugated composite plate was completed in 19.2 s. The period length increased with time. In 0–8.64 s, the selected composite plate region did not enter the straightening roll, so that the period length change was 0 and remained 10 mm. Afterwards, the composite plate was affected by roller to move along the straightening direction. In 8.16–9.6 s, the plate was elongated by a large reduction and the period length became 10.06 mm. In 9.6–10.56 s, the plate was moved from the fourth roll to the sixth roll. The period length increased to 10.11 mm, but the increasing trend declined. In 10.56–12 s, the plate was moved to the eighth roll. The period length increased to 10.14 mm and the increasing trend was further reduced. In 12–13.44 s, the plate was moved to the tenth roll. The period length was elongated for the last time, and it was 10.15 mm. Since then, the period length showed no change.

In Figure 8, the amplitude characteristic curve with straightening time is shown. In 0–8.64 s, the amplitude showed no change and remained 0.2 mm because the selected region did not enter the straightening roll. In 8.16–9.6 s, the amplitude became 0.206 mm because it is affected by the roll when the selected region moved along the straightening direction. However, the amplitude gradually decreased in the subsequent straightening. After 12 s, the amplitude changed in a wavy manner around 0.203 mm, which should be attributed to the elastic deformation of the bonding layer in the normal direction.

The yield strength of Mg and Al alloy is closely related to temperature. The yield strengths of AZ31B Mg and 5052 Al at 25 °C to 375 °C are shown in Figure 9. The straightening temperatures of 100 °C, 150 °C, 200 °C, 250 °C, 300 °C and 350 °C were selected for simulation, and the yield strength curve with different temperatures are shown in Figure 10.

In Figure 9, the yield strength decreased with the temperature. The trend of yield strength changed greatly between 200 °C and 300 °C. Above 300 °C, the trend was stable. Therefore, the appropriate straightening temperature was helpful in improving bonding properties. However, the influence of interfacial microcracks and surface cracks caused by the increase in interfacial stress on the quality of composite plates should be considered.

In Figure 10, the changes of period length and amplitude in the bonding layer of Mg/Al corrugated composite plate were studied at different temperatures. The amplitude and period length both increased along with temperature. In the range of 200–300 °C, the growth rate of amplitude and period length increased rapidly. Above 300 °C, the increase in morphology interface area slowed down and the stress on the plate was beyond yield strength, which could cause the plate to be cracked or fractured. Therefore, the suitable straightening temperature was 200–300 °C.

Different speeds, roll distances and roll diameters were set for straightening to explore the influence of straightener parameters on interface. The changes of morphology interface were discussed.

The roll-speed-morphology profiles of interface are explored in Figure 11. The roll speed values of 1 rad/s, 1.5 rad/s, 2 rad/s, 2.5 rad/s and 3 rad/s were set, respectively, for the straightening simulation. The amplitude and period length both decreased along with roll speed. According to Equation (8), the increase in roll speed meant that the bending moment by roll decreased when other conditions remained unchanged. It was the reason that the reduction of amplitude was significantly greater than that of period length.

The roll-distance-morphology profiles of interface are explored in Figure 12. The roll diameters of 90 mm, 95 mm, 100 mm, 105 mm and 110 mm were set, respectively, for the straightening simulation. The amplitude and period length both decreased along with the increase in roll distance. When roll distance increased, bite angle α declined, so the bending moment by the roll decreased. Moreover, the material deformation along the normal direction was greatly affected by the roll distance. Therefore, the reduction of amplitude was significantly greater than that of period length.

The roll-diameter-morphology profiles of interface are explored in Figure 13. The roll diameter of 85 mm, 90 mm, 95 mm, 100 mm and 105 mm were set, respectively, for the straightening simulation. In Figure 13, the change of roll diameter showed little influence on the interface morphology. The variation range of amplitude was from 0.2061 mm to 0.2067 mm and the variation range of period length was from 10.148 mm to 10.154 mm. The change of them was less than 0.3%. This could be measured by simulation software, but it was difficult to measure effectively in the actual straightening process. Therefore, the influence between roll diameter and interface morphology could be negligible.

The straightening parameters affect the quality of the straightened composite plate. In this work, the overall inclined straightening scheme was adopted. According to Equations (9) and (10), three different straightening process parameters were defined, and they are shown in Table 5. The straightening process parameters–morphology profiles of interface are explored in Figure 14.

In Figure 14, the interface morphology was affected by straightening process parameters. In particular, the amplitude and period length gradually decreased with the reduction of inlet roll. Therefore, the effective area of corrugated interface decreased, and bonding strength would decrease when the bonding strength of the unit area was certain. However, the change trend of amplitude was greater than that of period length.

The residual stress of plate surface after straightening is shown in Figure 15. In Figure 15, the residual stress range of the plate with different projects was small and it was below 30 MPa, which would not affect the subsequent use. However, the maximum residual stress of the plate decreased with the reduction of the second roller and the uniformity of the residual stress distribution was affected by the tenth roller. The residual stress of the Al layer was greater than that of the Mg layer. After the straightening of project 1, the surface of the plate was flattest.

Therefore, it is essential to optimize the straightening process to achieve the balance between the bonding interface and the residual stress.

## 6. Discussion

Designed factorial experiments were employed in order to quantify the influence and effects of a number of variable process parameters, in the eleven-roll straightener, on the interface characteristics of the Mg/Al corrugated composite plate.

In straightening, the amplitude and period length increased with time, and the increase was mainly in the first straightening unit. The amplitude increased by 1.1% and the period length increased by 3%. Combined with the results of different projects, when the reduction of the second roll decreased, the amplitude and period length also decreased. According to reference [10], the advantage of corrugated rolling is that it increased the interface area and improved the atomic penetration; thus, the bonding strength of the composite plate increased. Therefore, the reduction of the inlet roll should be increased as much as possible while ensuring the straightening effect and uniform stress distribution. In addition, the straightening temperature not only determined the design of the project, but also affected the interface morphology of the composite plate. Because of the sensitivity of Mg and Al to temperature, the relationship between temperature, straightening effect and interface morphology needed to be balanced. On the one hand, the increase in temperature could increase the interface area to improve bonding strength. On the other hand, to increase the temperature would reduce the straightening effect and increase the possibility of cracks in the plate. Combined with the results of Figure 10 and reference [32], the appropriate straightening temperature was 250 °C. Above 250 °C, the change trend of the combined area slowed down and the straightening effect was poor due to a yield strength of less than 75 MPa.

According to the equation of straightener parameters, the size of roll length determined the stress distribution in the width direction. It depended on the width of the plate and could not affect the interface morphology. The straightener parameters determined the bending moment and the shear strength necessary to change the interface morphology. The interface area increased with the raising of roll diameter and decreased with the diminishing of roll distance and roll speed. The fast roll speed would decrease the surface quality of the plate and the slow roll speed would reduce straightening efficiency. In Figure 11, the decreasing trend of the morphology area was slow, with 1 rad/s to 2 rad/s, so the best roll speed was 2 rad/s. Due to the decreasing roll distance, the bending moment was diminished to cause the reduction of the morphology area. However, increasing the roll distance could not only reduce the pressure of each roll, but also could increase the biting conditions. The effect of roll diameter and roll distance was opposite, but the principle was consistent. So, 100 mm was considered the best roll distance and 95 mm was considered the best roll diameter in this working condition.

## 7. Conclusions

The interface properties of Mg/Al corrugated rolled composite plate was studied via experiment. According to the result, the interface morphology was expressed by the amplitude and period length of the equation. Moreover, the Mg/Al corrugated rolled composite was built in Abaqus software. After straightening, the trend of interface morphology was discussed. The reduction of the second roll played a role on the interface morphology. Compared the other roller, the amplitude of the second roll increased by 1.1% and the period length by the second roll increased by 3%.By analyzing the equation of straightener parameters, shear deformation and bending moment had an important impact on the deformation of the straightening direction. Therefore, in straightening, different sizes of roll distance, roll diameter and roll speed were built in Abaqus. The results showed that increasing amplitude and period length was based on roll distance and roll diameter, and decreasing amplitude and period length was based on roll speed. The fit straightener parameters were selected.The different straightening process projects were designed by elastoplastic mechanics and were applied in Abaqus. The relationship between residual stress and project was explored. Finally, the fit straightener parameters were selected considering interface morphology and residual stress.The Mg/Al corrugated composite plate had been exploited in many fields, but after rolling, shape defect still existed. This work could provide a theoretical basis for improving the straightening accuracy of Mg/Al corrugated composite plate considering interface morphology. It could be applied in theoretical calculation and practical processing.

## Figures and Tables

**Figure 1 materials-15-04383-f001:**
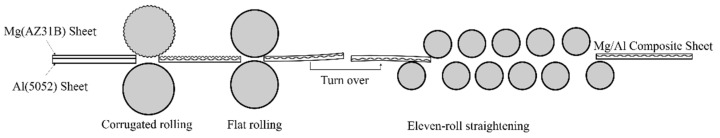
Schematic diagram of rolling-straightening process of Mg/Al corrugated composite plate.

**Figure 2 materials-15-04383-f002:**
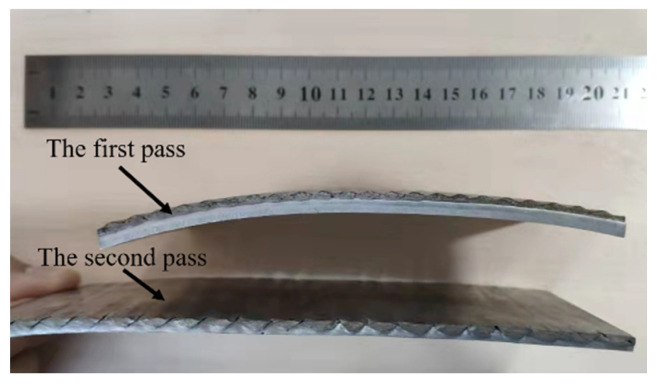
Schematic diagram of Mg/Al corrugated composite plate after rolling in different passes.

**Figure 3 materials-15-04383-f003:**
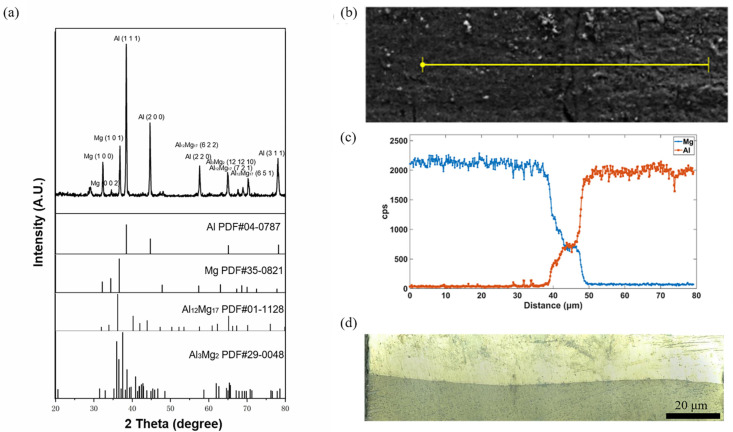
Interface properties of Mg/Al corrugated composite plate: (**a**) XRD patterns; (**b**) position of line scan by SEM; (**c**) content of Mg and Al by EDS; (**d**) interface morphology.

**Figure 4 materials-15-04383-f004:**
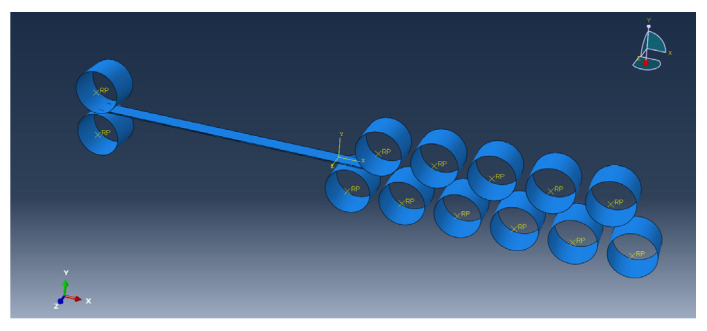
Schematic diagram of 11-roll straightening model.

**Figure 5 materials-15-04383-f005:**
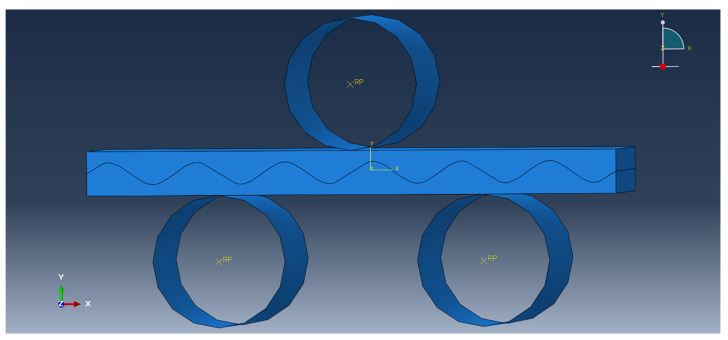
Schematic diagram of single straightening unit of composite plate.

**Figure 6 materials-15-04383-f006:**
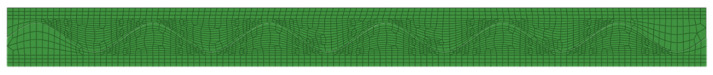
Grid division.

**Figure 7 materials-15-04383-f007:**
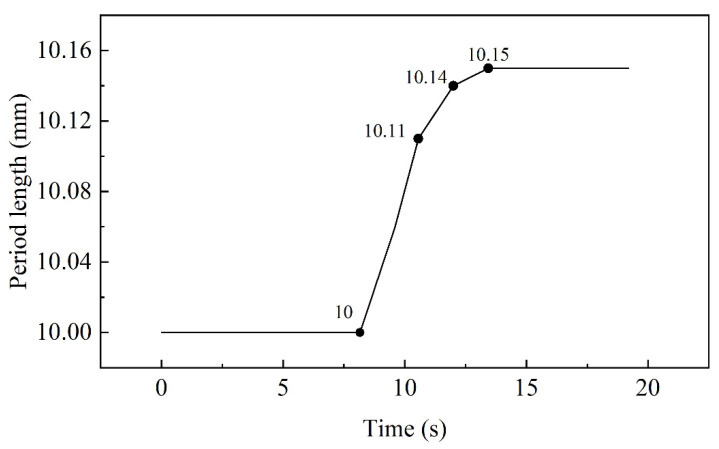
The period length-straightening time profiles of interface.

**Figure 8 materials-15-04383-f008:**
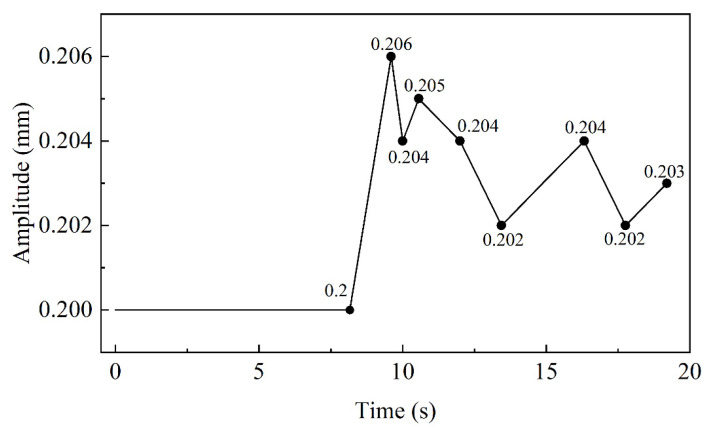
The amplitude-straightening time profiles of interface.

**Figure 9 materials-15-04383-f009:**
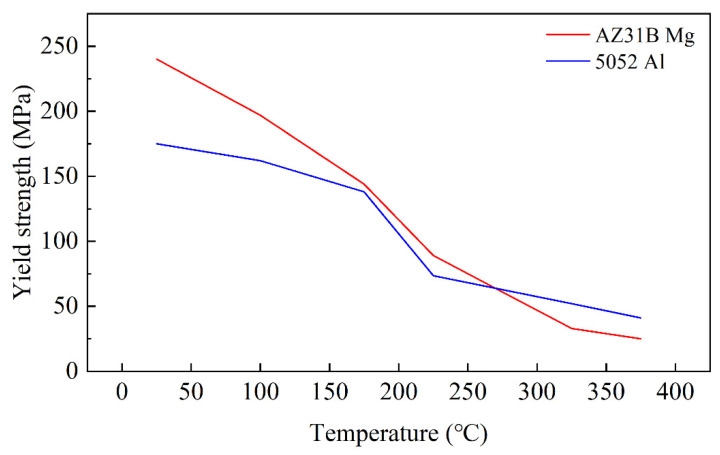
The yield strength curve with different temperatures.

**Figure 10 materials-15-04383-f010:**
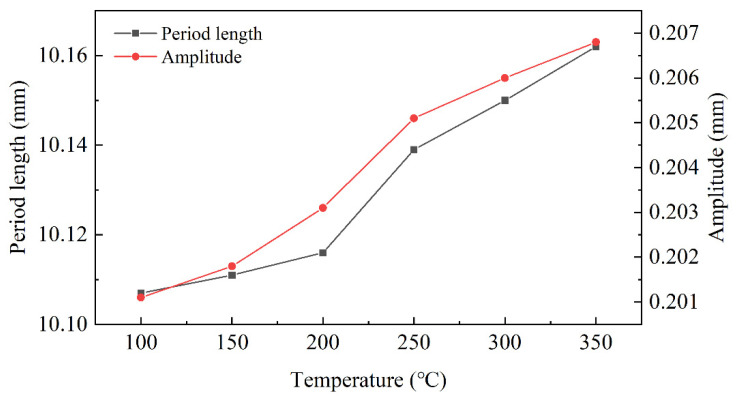
The morphology curve with different temperatures.

**Figure 11 materials-15-04383-f011:**
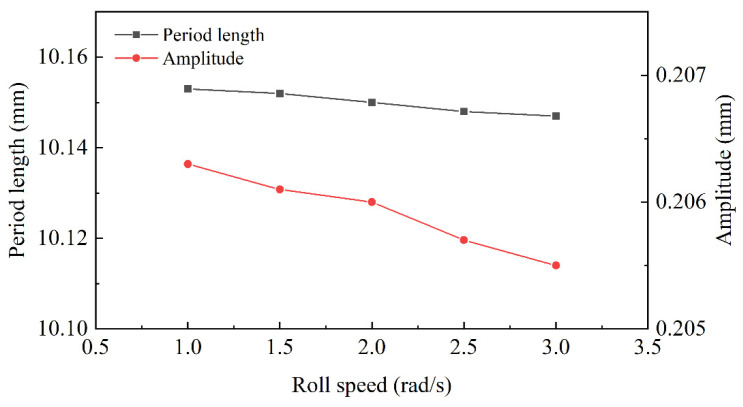
The morphology curve with different roll speeds.

**Figure 12 materials-15-04383-f012:**
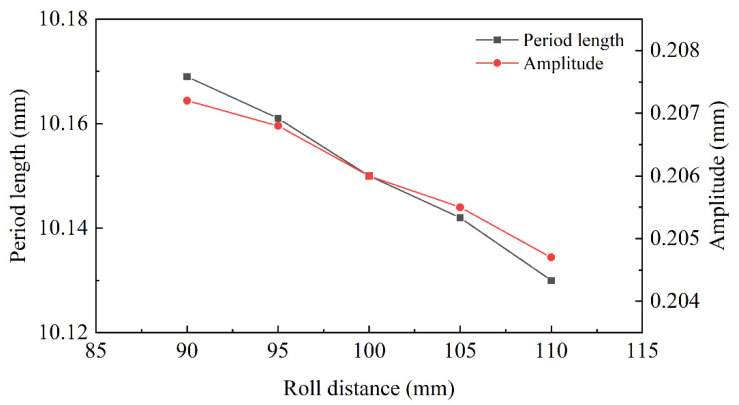
The morphology curve with different roll distances.

**Figure 13 materials-15-04383-f013:**
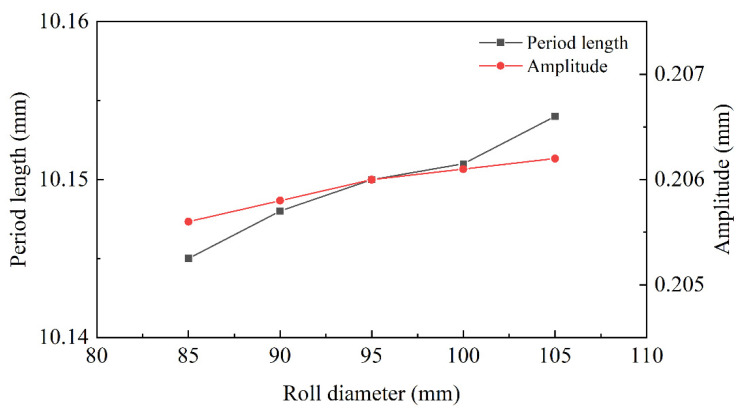
The morphology curve with different roll diameters.

**Figure 14 materials-15-04383-f014:**
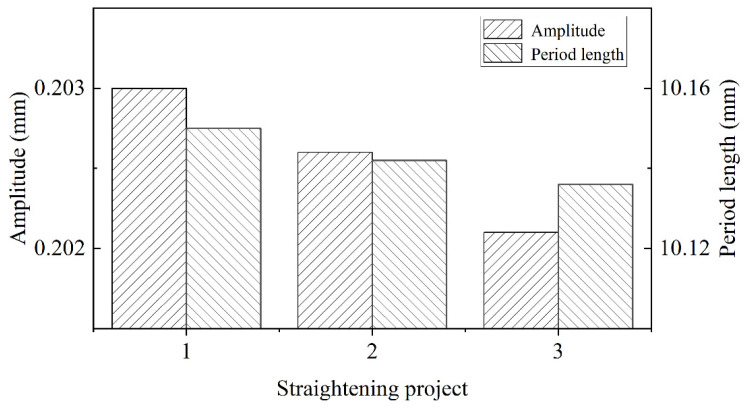
The morphology curve with different straightening projects.

**Figure 15 materials-15-04383-f015:**
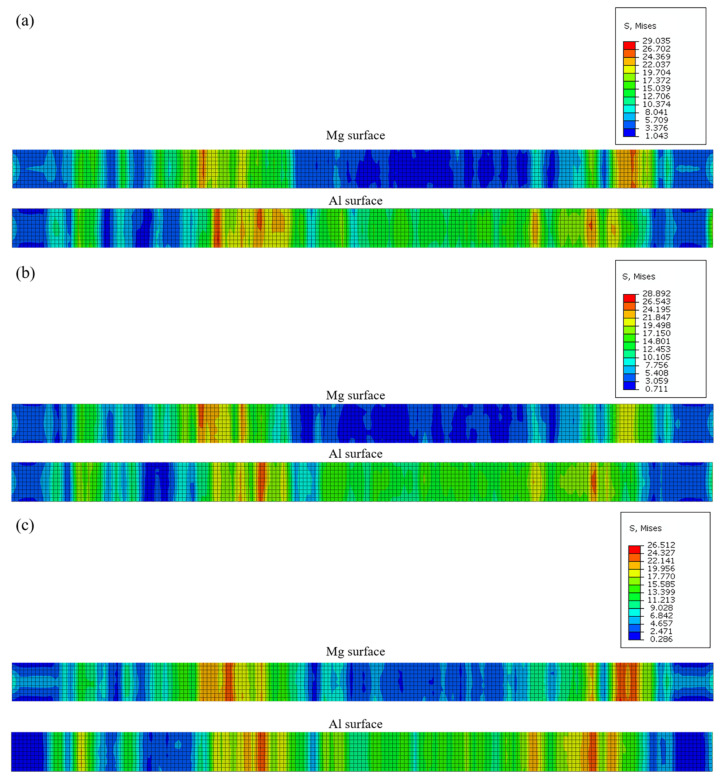
The residual stress of plate surface with different straightening projects: (**a**) straightening project 1; (**b**) straightening project 2; (**c**) straightening project 3.

**Table 1 materials-15-04383-t001:** Rolling parameters of Mg/Al corrugated composite plate.

Roll Diameter/mm	Rolling Temperature	Rolling Speed	Reduction Rate	Tempering Time	Roll Parameters
First pass	400 °C	0.98 rad/s	32%	30 min	150 mm × 150 mm y = (75 + 0.55sin(100 × t)) × cost X = (75 + 0.55sin(100 × t)) × sint
Second pass	400 °C	1.47 rad/s	28%	15 min	250 mm × 300 mm

**Table 2 materials-15-04383-t002:** General parameters of straightener.

Roll Diameter/mm	Roll Length/mm	Roll Distance/mm	Poisson Ratio	Roll Speed/rad/s	Kinetic Friction
95	205	100	0.3	2	0.25

**Table 3 materials-15-04383-t003:** Reduction of upper straightening roll.

Roll Number	2	4	6	8	10
Reduction/mm	0.94	0.71	0.49	0.24	0

**Table 4 materials-15-04383-t004:** General parameters of composite materials.

Materials	Yield Strength $\sigma_{s}$/MPa	Elasticity Modulus E/GPa	Density ρ/(kg∙m^−3^)	Poisson Ratio μ
Mg	51.4	257	1.77	0.3
Al_12_Mg_17_	52	270	1.8	0.3
Al_3_Mg_2_	42.5	324	2.5	0.3
Al	53.4	376	2.72	0.3

**Table 5 materials-15-04383-t005:** Straightening process parameters.

Straightening Program	2	4	6	8	10
1	0.94 mm	0.71 mm	0.49 mm	0.24 mm	0 mm
2	0.87 mm	0.64 mm	0.41 mm	0.28 mm	0.05 mm
3	0.78 mm	0.56 mm	0.35 mm	0.13 mm	0.08 mm

## Data Availability

Not applicable.

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
