# Peer review of "Investigation on the Interface Morphology of Mg/Al Corrugated Composite Plate in the Straightening Process"

_materials, 2022, doi:10.3390/ma15134383_

Round 1
Reviewer 1 Report
What is the novelty of work? Are you sure cannot find any papers in the literature close to the presented work? In a table please present the title of the published paper in this field
1. the system has been considered to be fully symmetric. Why? A justification required.
1. the geometric nonlinearities have been neglected. Why? It has been shown in the literature that they can be very important in the behavior of the structures. A justification to be given.
contributions of the paper should be clearly outlined in the last paragraph of the Introduction section to justify the motivation for this study. it is not very clear for a general reader.
double-check the grammar of the text.
All symbols and parameters in equations should be defined (it is necessary)
conclusion section should be expanded and enlarged for more elaboration and clarification
quality of figures should be improved. They are not acceptable for publication
The literature review on shear deformation effect is very poor. Authors must improve this point
Author Response
The authors greatly appreciate the reviewer for the precious constructive comments and suggestions on our manuscript. The manuscript has been revised according to the comments. The amendments to the manuscript are described, where we include the replies to the questions raised by the reviewer.

Reviewer 2 Report
Please make the specified updates.

Author Response

(The authors gave the same response as above.)

Reviewer 3 Report
The first 3 sentences in abstract are vague and repetitive -please draft a more structured abstract !
This is quite large range “200 ℃ –300 ℃” !
Please highlights the industrial impact of this work !
It is requested to define all the acronyms before their first appearance in text
Your manuscript is little bit confusing as in abstract you have suggested everything was determined by numerical calculations, however in introduction you said that you have used XRD and EDS to determine the interface !
The scientific contribution is not well highlighted
“composite interface bonding layer was divided into Al12Mg17 layer and Al3Mg2” not clear how was made this in abaqus ! – please describe it carefully
Your method and material should be refined as not clear each step involved for this research
Figure 3 d requires a scale bar
How you have determined the properties of compound “Al12Mg17” and “Al3Mg2” which were indicated in Table 4 ?
Figure 6 should be rescaled
The static coefficient do not have actually a physical meaning and seems quite low,
In which basis you have determined the temperature used ? cause it is not clear their physical meaning
The legend of figure 15 is not well visible
There is required a section of discussion
Which version of abaqus was used ?
Author Response

(The authors gave the same response as above.)

Round 2
Reviewer 2 Report
Thank you for the arrangement. Congratulations on your extensive work. I would like to inform you that it will be published with the final version. Of course, the final decision belongs to the journal editors. have a nice day.
Reviewer 3 Report
[